# State Observability through Prior Knowledge: Analysis of the Height Map Prior for Track Cycling [note 1]

**DOI:** 10.3390/s20092438

**Published:** 2020-04-25

**Authors:** Tom L. Koller, Udo Frese

**Affiliations:** Multi-Sensor Interactive Systems Group, University of Bremen, 28359 Bremen, Germany; ufrese@uni-bremen.de

**Keywords:** observability analysis, IMU, INS, prior knowledge, C = context knowledge, vehicle constraint, pose estimation, tracking

## Abstract

Inertial navigation systems suffer from unbounded errors in the position and orientation estimates. This drift can be corrected by applying prior knowledge, instead of using exteroceptive sensors. We want to show that the use of prior knowledge can yield full observability of the position and orientation. A previous study showed that track cyclers can be tracked drift-free with an IMU as the only sensor and the knowledge that the bike drives on the track. In this paper, we analyze the observability of the pose in the experiment we conducted. Furthermore, we improve the pose estimation of the previous study. The observability is analyzed by testing the weak observability criterion with a Jacobian rank test. The improved estimator is presented and evaluated on a dataset with three 60-round trials (10 km each). The average RMS is 1.08 m and the estimate is drift-free. The observability analysis reveals that the system can gain complete observability in the curves and observability of the orientation on the straight parts of the race track.

## 1. Introduction

For short time periods, the pose (position and orientation) of an object can be estimated with an inertial measurement unit (IMU). The pose is estimated by integrating the acceleration and angular rate measurements. This method accumulates the measurement errors of the IMU, wherefore the estimate drifts over time. The drift can be corrected by fusing the IMU with exteroceptive sensors; e.g., a global navigation satellite system (GNSS).

In indoor environments, GNSS is unavailable. Custom radio emitters or a building’s WiFi can be used instead [1]. Fusing with an external sensor seems essential. Surprisingly, several pedestrian tracking systems achieve drift-free estimates with an IMU alone [2,3,4]. The systems fuse IMU measurements with prior knowledge of the environment and specific gait motion dynamics, instead of using an additional sensor.

Fusing IMU measurements with prior knowledge is a powerful concept with which to gain drift-free estimates [5]. The knowledge allows one to observe otherwise unobservable states. These approaches have the advantage that they do not require additional sensor hardware in the buildings. Furthermore, most users already carry the necessary IMU in their smartphone [2].

Motivated by the success of prior knowledge in pedestrian tracking [2,6,7] and vehicle tracking [8,9,10] we conducted an experiment to track cyclists (see Figure 1) with an IMU as the only sensor [11]. The estimated trajectory was drift-free in three 20 min trials with an RMS of about 1.15 m. This paper extends the experimental work [11] with an observability analysis and an improved estimator with an RMS of about 1.08 m.

To achieve drift-free estimates, the IMU measurements were combined with two priors. The first is the wheeled vehicle constraint of [8], which we call the forward velocity prior. It states that a wheeled vehicle has approximately zero velocity perpendicular to its forward direction. This prior allows one to observe the forward velocity from IMU measurements if the vehicle is driving on a curve. Additionally, IMU biases are observable [12]. Since the bike track contained two 180° curves (see Figure 1), the velocity and biases became periodically observable with the prior.

The second prior is the height map prior, which states that the bikes have to stay on the track. Several works use maps to improve the state estimate. The map can have different forms; e.g., route maps [10,13,14] with the paths that can be taken or building plans [3] with impassable walls. With a terrain map, a vehicle can be localized without a GNSS [9]. Similarly, the track of the race (see Figure 1) can be modeled as a height or terrain map. In contrast to the forward velocity prior, the observability with a map prior is unknown.

The focus of this extended version is to analyze whether or not the proposed system can observe the full pose of the bike. Thus, we analyze the observability properties of map priors in conjunction with the forward velocity prior. The goal of the analysis is to understand the error behavior of the system and to evaluate the qualification of the system to track the track cycler.

Informally, we mean by observable that the pose estimate does not drift away from the real pose. This is important in practice, since applications such as visualization of a race require low errors of pose estimates. While the error bounds may be evaluated by excessive experiments, an analytical analysis of the observability reveals weak spots of the measurement system. This allows one to specifically test error cases of the system and to predict the overall error behavior.

The contribution of this extended paper is the observability analysis of the height map prior for wheeled vehicles. Different surface types are analyzed to reflect the two major parts of the race track, the curves and the straight parts. In addition, an improved version of the original trajectory estimator [11] is presented, which takes account of the bike dynamics [15]. The estimator is evaluated on the original dataset.

## 2. Materials and Methods

### 2.1. Knowledge-Based Optimization Problem for Estimation

In track cycling, bikers race on a track. There are several game modes with varying strategies [16]. In our setup, the bikes are equipped with wireless IMUs, as shown in Figure 2.

The trajectory is estimated by finding the trajectory with the maximum a-posteriori probability given the measurements of the main frame IMU, the dynamic model and the prior knowledge. We formulate the state estimation as a minimization problem and solve it by using the ceres least squares solver [17]. The solver works offline and optimizes the whole problem at once.

The trajectory can be modeled as a sequence of *n* states $x_{k}$, 0<k≤n (see Figure 3) connected by the dynamic model:(1)$$x_{k+1}=g_{k}\left(x_{k}\right)$$

Instead of using the IMU input for the dynamic update, the state does contain every bit of information needed for the dynamic model. The inputs are used as a constraint on the respective state. The prior knowledge constrains every state of the trial.

Each state $x_{k}$ consists of the position ${\vec{p}}_{W}$, velocity ${\vec{v}}_{W}$ and acceleration ${\vec{a}}_{W}$ in world frame; the rotation from body to world $Q_{W}^{B}$, the angular rate ${\vec{\omega }}_{B}$ and acceleration ${\vec{\omega }}_{B}^{\prime }$; and two-part biases of the accelerometer ${\vec{b}}_{a}={\vec{b}}_{a1}+{\vec{b}}_{a2}$ and the gyrometer ${\vec{b}}_{\omega }={\vec{b}}_{\omega 1}+{\vec{b}}_{\omega 2}$. The biases are modeled as the sums of two biases following the model in [18], which claims to be more precise than the standard bias model. The components always refer to $x_{k}$, wherefore we can omit the *k*-indices for readability:(2)$$x_{k}={\left(\begin{array}{cccccccccc} {\vec{p}}_{W} & {\vec{v}}_{W} & {\vec{a}}_{W} & Q_{W}^{B} & {\vec{\omega }}_{B} & {\vec{\omega }}_{B}^{\prime } & {\vec{b}}_{a1} & {\vec{b}}_{a2} & {\vec{b}}_{\omega 1} & {\vec{b}}_{\omega 2} \end{array}\right)}^{T}$$

The orientation $Q_{W}^{B}$ is modeled as an Euler–Rodriguez rotation matrix. Usually, the acceleration and angular rate are modeled as inputs instead of states. We chose to add them to the state to apply prior knowledge to them. Originally, we defined the point on the ground under the main frame IMU as the body frame (see Figure 2), instead of the IMU pose, because the chosen prior knowledge does not hold for the IMU pose. Taking the bike dynamics into account [15], the forward velocity prior does only hold at the center of the rear wheel. At other points, side velocities appear when the bike drives a curve. Thus, we changed the body frame to the position on the ground under the center of the rear wheel.

The complete parameter space *X* of the optimization problem consists of the states at all steps:(3)$$X=\{x_{1},\cdots ,x_{n}\}$$

We want to find the most likely sequence of states given the inputs ${\vec{ac}}_{k}$ and ${\vec{gy}}_{k}$, the input functions $a_{k}\left(x_{k}\right)$ and $\omega_{k}\left(x_{k}\right)$, the dynamic model $g_{k}\left(x_{k}\right)$ and our prior knowledge; i.e., the height map $h_{k}\left(x_{k}\right)$ and the forward velocity prior $fv_{k}\left(x_{k}\right)$. This can be calculated by solving the minimization problem:(4)$$\hat{X}=\underset{X}{\arg\;\min}\sum_{k=1}^{n}\left[\underset{\mathrm{gyrometer}\;\mathrm{input}}{\underset{︸}{\left|\right|\omega_{k}\left(x_{k}\right)-{\vec{gy}}_{k}{\left|\right|}_{\Sigma_{gy}}^{2}}}+\underset{\mathrm{accelerometer}\;\mathrm{input}}{\underset{︸}{\left|\right|a_{k}\left(x_{k}\right)-{\vec{ac}}_{k}{\left|\right|}_{\Sigma_{ac}}^{2}}}+\underset{\mathrm{dynamic}\;\mathrm{model}}{\underset{︸}{\left|\right|g_{k}\left(x_{k}\right)⊟x_{k+1}{\left|\right|}_{\Sigma_{g}}^{2}}}+\underset{\mathrm{forward}\;\mathrm{velocity}\;\mathrm{prior}}{\underset{︸}{||fv_{k}\left(x_{k}\right){\left|\right|}_{\Sigma_{fv}}^{2}}}+\underset{\mathrm{heightmap}\;\mathrm{prior}}{\underset{︸}{\left|\right|h_{k}\left(x_{k}\right){\left|\right|}_{\Sigma_{h}}^{2}}}\right]$$
where $||v-\hat{v}{\left|\right|}_{\Sigma }^{2}$ is the Mahalanobis distance. The ⊟ is a normal minus operation, despite the fact that—for rotation matrices—it returns the difference as a rotation vector; i.e., a unit axis scaled by an angle [19]:(5)$$x^{*}⊟x=\left\{{\vec{p}}_{W}^{\;*}-{\vec{p}}_{W},\cdots ,\overset{︷}{\log({Q_{W}^{B}}^{T}Q_{W}^{B*})},{\vec{\omega }}_{B}^{*}-{\vec{\omega }}_{B},\cdots \right\}$$

Its purpose is to handle the manifold structure of $Q_{W}^{B}$ as proposed in [19].

The minimization problem is formulated by simply adding up information about the states. The first pool of information added is the gyrometer input at each time step. Each state has to match the measured angular rate. With $\omega_{k}\left(x_{k}\right)$, we calculate a prediction of the angular rate and compare it with the measurement. By imposing the difference of the prediction and the measurement as an error, the optimizer will adapt each state $x_{k}$ until the prediction and the measurement coincide as good as possible. This is similar to the measurement step in a Kalman filter.

The prediction of the angular rate can be calculated by:(6)$$\omega_{k}\left(x_{k}\right)=Q_{I}^{B}*{\vec{\omega }}_{B}+{\vec{b}}_{\omega }$$
where the calibrated rotation matrix $Q_{I}^{B}$ transforms ${\vec{\omega }}_{B}$ from body to IMU frame. The ∗ denotes matrix multiplication.

The second piece of information that we have about the states is the accelerometer input. Similar to the gyrometer input, the predicted accelerations $a_{k}\left(x_{k}\right)$ and the acceleration measurements ${\vec{ac}}_{k}$ have to match. Since the IMU is not at the origin of the body frame, the IMU measures accelerations induced by rotational movements. Additionally, the acceleration in the state is gravity free. Those accelerations are taken into account by the prediction function:(7)$$\begin{array}{cc} a_{k}\left(x_{k}\right) & =Q_{I}^{B}*{\left(Q_{W}^{B}\right)}^{T}*\left({\vec{a}}_{W}-{\vec{g}}_{W}\right)\\ \; & +Q_{I}^{B}*\left({\left[{\vec{\omega }}_{B}^{\prime }\right]}_{\times }+{\left[{\vec{\omega }}_{B}\right]}_{\times }^{2}\right)*{\vec{t}}_{I}+{\vec{b}}_{a} \end{array}$$
where ${\vec{g}}_{W}$ is the gravity vector and ${\left[\cdots \right]}_{\times }$ forms a skew symmetric matrix out of the given vector. ${\vec{t}}_{I}$ is the calibrated offset between IMU and body frame.

The dynamic model $g_{k}\left(x_{k}\right)$ connects the states $x_{k}$ and $x_{k+1}$ over time. The predicted next state $g_{k}\left(x_{k}\right)$ has to match the next state $x_{k+1}$. The dynamic model is based on the classic inertial navigation system (INS) state space model (Section 3.7.1 of [20]):(8)$$\begin{array}{cc} g_{k}\left(x_{k}\right) & =\left(\begin{array}{c} {\vec{p}}_{W}+\Delta t\cdot {\vec{v}}_{W}+\frac{\Delta t^{2}}{2}\cdot {\vec{a}}_{W}\\ {\vec{v}}_{w}+\Delta t\cdot {\vec{a}}_{W}\\ {\vec{a}}_{W}\\ Q_{W}^{B}*\exp\left(\Delta t\cdot {\vec{\omega }}_{B}\right)\\ {\vec{\omega }}_{B}+\Delta t\cdot {\vec{\omega }}_{B}^{\prime }\\ {\vec{\omega }}_{B}^{\prime }\\ {\vec{b}}_{a1}\cdot \left(1-\frac{\Delta t}{T_{cor1}}\right)\\ {\vec{b}}_{a2}\cdot \left(1-\frac{\Delta t}{T_{cor2}}\right)\\ {\vec{b}}_{\omega 1}\\ {\vec{b}}_{\omega 2}\cdot \left(1-\frac{\Delta t}{T_{cor3}}\right) \end{array}\right)\\ T_{cor1} & =1000\;s,\;\;\;\;T_{cor2}=T_{cor3}=2000\;s \end{array}$$
where Δt is the time difference. The accelerometer bias is modeled as the sum of two exponential auto correlated random walk functions with decorrelation constants $T_{cor1}$ and $T_{cor2}$. The gyrometer bias is modeled as the sum of a constant and an exponential auto correlated random walk function with decorrelation constant $T_{cor3}$. exp(⋯) is the Euler–Rodriguez Formula [19]. Cartesian and angular acceleration stay constant in the dynamic model, but do not contribute to the error calculation.

In contrast to the other information, the prior knowledge sources do not have a measurement to compare. However, prior knowledge often has a target value which is either constant or state-dependent itself. For example, the forward velocity prior can be formulated as [8]:(9)$$fv_{k}\left(x_{k}\right)={\left(Q_{W}^{B}\right)}^{T}*{\vec{v}}_{W}-\left(\begin{array}{c} {}*\\ 0\\ 0 \end{array}\right)=\left(\begin{array}{ccc} 0 & 1 & 0\\ 0 & 0 & 1 \end{array}\right)*{\left(Q_{W}^{B}\right)}^{T}*{\vec{v}}_{W}$$
where the target for the y and z dimensions of the body velocity is 0. The asterisk is a wildcard, which allows arbitrary velocity in forward direction. If a measurement of the x dimension is available, e.g., by odometry, it can be used instead. In our model, the x dimension is unused.

The height-map prior states that the wheels of the bike stay at the track. We impose this constraint on the rear wheel only. This means that the distance between the position ${\vec{p}}_{W}$ and the closest point on the track has to be 0, or otherwise stated:(10)$$h_{k}\left(x_{k}\right)={\vec{p}}_{W}-\underset{{\vec{c}}_{W}\in Track}{\arg\;\min}\left|\left|{\vec{p}}_{W}-{\vec{c}}_{W}\right|\right|$$
where Track is the set of all points on the track. In this case, the prediction ${\vec{p}}_{W}$ is trivial but the target value is state dependent.

In this manner, additional information can be added simply by adding it to (Equation 4). The optimizer will incorporate it weighted by the covariance of the information.

The covariances are crucial tuning parameters of the model. Neither the forward velocity prior, nor the height-map prior hold exactly. Side slip can occur, which results in a side velocity. The height map itself is imperfect. Furthermore, the bike may bounce on surface irregularities. Those imprecisions of the knowledge are modeled in the covariance of the measurement equations. If the knowledge is violated more than expected, the estimator’s performance will be reduced.

#### 2.1.1. Initial Guess

The convergence of a least squares solver depends on the initial guess of the states. Especially in the presence of state dependent constraints, the solver may stay in unsuitable local minima. We designed a method based on prior knowledge to provide an initial guess for the least squares solver. This method is again based on another, but more simple model.

The initial guess method exploits the basic form of the track. In principle, the track consists of two straight lines connected with two curves (see Figure 4). As a vague model of the track, we assume that the bikers follow the 1D line. This model requires big noise in lateral direction and for orientation. However, the error of the 1D model is bounded. In lateral direction, the bikers are forced to stay on the track, which has a width of 6 m. Hence, they can not drive far away from the 1D line. Their yaw is also constrained to follow the 1D track approximately, because they have to take the track counter clockwise.

The 1D formulation of the track has the advantage that we can apply Theorem 1 from [5], which states that a 1D system is observable from gyrometer measurements alone if the rotation axis changes. At the entries and exits of the curves, the rotation axis changes. Hence, they can be detected.

Between the entries and exits of the curves, dead reckoning is required. To reduce the drift, we can apply the forward velocity prior [8]. It makes the forward velocity observable. Hence, it can be estimated drift-free.

The initial guess is calculated by an unscented Kalman filter (UKF). Each state $x_{k}$ consists of:(11)$$\begin{array}{cc} x_{k} & ={\left(\begin{array}{cccccc} \lambda & \dot{\lambda } & Q_{W}^{B} & {\vec{\omega }}_{B} & {\vec{v}}_{W} & {\vec{a}}_{W} \end{array}\right)}^{T} \end{array}$$
where λ is the position on the 1D line, $\dot{\lambda }$ the speed on the line and the rest is defined as in (Equation 2). The dynamic model is similar defined as Equation (Equation 8) by:(12)$$\begin{array}{cc} g(x_{k}) & =\left(\begin{array}{c} \lambda +\Delta t\cdot \dot{\lambda }\\ \dot{\lambda }\\ Q_{W}^{B}*\exp\left(\Delta t\cdot {\vec{\omega }}_{B}\right)\\ {\vec{\omega }}_{B}\\ {\vec{v}}_{W}+\Delta t\cdot {\vec{a}}_{W}\\ {\vec{a}}_{W} \end{array}\right) \end{array}$$

We formulated the main model by taking the difference of the measurement and a prediction function of the measurement as error. The same prediction function can be used in the measurement step of the UKF to apply the same information. In the case of prior knowledge, 0 is used as the target value. This is called a pseudo measurement [21]. In this manner, Equations (Equation 6) and (Equation 7) are used to incorporate the IMU measurements. Equation (Equation 9) is used to apply the forward velocity prior.

In the 1D model, the bike follows the line only. Hence, we have to set $\dot{\lambda }$ to the norm of the velocity. This is done as a perfect measurement using the prediction function:(13)$$\begin{array}{cc} {\dot{\lambda }}_{p}\left(x_{k}\right) & =\left|{\vec{v}}_{W}\right|-\dot{\lambda } \end{array}$$

Since the biker has to follow the track’s direction, the angular rate has to be the direction change of the track approximately. In other words, the biker is likely to be in a curve if the gyrometer measures a nonzero angular rate. The prediction function is:(14)$$\begin{array}{cc} \omega_{p}\left(x_{k}\right) & ={\vec{\omega }}_{B}-C\left(\lambda \right)\cdot \dot{\lambda } \end{array}$$
where C(λ) is a function which returns the curvature of the 1D line at a given λ. This prior allows one to observe the entry and exit points of the curves.

The UKF evaluates the probability distribution at different sigma points. If the covariance of the filter is underestimated, i.e., the UKF is too confident in the estimate, the distribution is evaluated poorly. Therefore, we apply high process noise, which allows the filter to correct implausible estimates. The high process noise models the vagueness of the 1D model.

The pseudo measurements are only useful on the curves of the track. During the straight parts, dead reckoning is performed implicitly. As a result, the covariance rises quickly on the straight segments and is low when the biker drives a curve. Hence, the estimate jumps whenever the bike enters a curve. The estimate jumps are smoothed with the unscented Rauch–Tung–Striebel smoother [22].

#### 2.1.2. Implementation Details

The main model has been implemented in ceres-solver, an extensive nonlinear least squares library [17]. It provides an automatic differentiation framework, which simplifies the use of nonlinear constraints; e.g., the prior knowledge. The local parameterization of ceres is used to adapt the rotation matrices without breaking them. Additionally, the rotation matrices are normalized every iteration.

Inputs, the dynamic model and the prior knowledge are implemented as cost functions. The covariances for the Mahalanobis distances are shown in Table A1 (initial guess) and Table A2 (main model) in Appendix A.

To calculate the penalty for leaving the track, the closest point on the track is required. We approximate the closest point by taking the closest point on the x-y plane; i.e.:(15)$$h_{k}\left(x_{k}\right)\approx {\vec{p}}_{W}-\underset{{\vec{c}}_{W}\in Track}{\arg\;\min}\left|\left(\begin{array}{ccc} 1 & 0 & 0\\ 0 & 1 & 0 \end{array}\right)*\left({\vec{p}}_{W}-{\vec{c}}_{W}\right)\right|$$

This enables us to precalculate the closest point for any x-y pair in a rectangle around the track and to store it in a 2D grid. Values between grid points are interpolated with a cubic spline using the algorithm of [23], which is already implemented in ceres. The smooth interpolation improves the gradient descent. Values outside the precalculated rectangle are extrapolated by taking the closest border point.

Since least square optimizers are sensitive to outliers, measurements exorbitantly outside the sensor range were removed from the dataset. They are treated as missing measurements. Missing measurements in the data are handled by assuming that they were 0 and drastically increasing the noise of each measurement. This allows the estimator to estimate the acceleration without allowing irregularly high accelerations. For further details, the complete implementation is available at GitHub (https://github.com/TomLKoller/ZaVI_TrackCycling).

### 2.2. Dataset

We recorded the dataset (Download at: http://www.informatik.uni-bremen.de/agebv/zavi.) at the track of the Sixdays Bremen (see Figure 1), an annual track cycling race. The track has a nominal length of 166.66 m and a width of 6 m. A height map (see Figure 5) was built based on the blueprints by [24].

We equipped the bike with four Xsens Awinda IMUs [25] at different links (see Figure 2). The placement was meant to allow us to estimate the joint angles of the bike using [26]. The crank IMU could be used to estimate the bike’s velocity via wheel turn odometry. The sensors were connected wirelessly and measure at 100 Hz. They buffer the sensor data for 10 s during connection losses. Still, around 5% of the measurements were lost due to a broken antenna.

The bikes were also equipped with turn rate sensors to measure velocity as a reference for the bikers only.

A tracking system that covers the complete track was not available. Thus, we used laser barriers to measure when the bikes passed a certain position.

We used six custom built laser barriers as the ground truth references. They triggered at 1024 Hz. The barriers were placed at the entries and exits of the curves (see Figure 5). Two barriers were perpendicular to the track. The other four were arranged in two crosses. In each trial, one laser barrier failed to record data.

The moment when the bike’s center line (see Figure 2) passes the laser barrier is used as the ground truth measurement. It is calculated as the center of the detected interval in each round. We approximate that this method has a random error around 5 ms. With a constant velocity assumption, the lateral position could be calculated at the crosses. With the known length of the bikes, the velocity could be estimated from the time the biker needs to pass each laser barrier.

The barrier clocks (2 ppm error) and the IMU base station (1 ppm error) were synchronized before each trial. The accumulated time error after each trial is negligible.

The alignment of the barriers with the 3D model is imperfect. Thus, an unknown bias of a few centimeters is introduced.

Two bikers participated in data recording using their own bikes. The dataset consists of two trials with Biker 1 and one with Biker 2. The bikers were instructed to ride in this sequence:Ride on the 166.66 m reference line.Stay in the 0.7 m corridor above the reference line.Stay in the lower track half.Use the whole track.Repeat (2) with constant velocity.Repeat (3) with constant velocity.Ride as you like.

All tasks except (7) were executed for 10 rounds. That resulted in a driving distance of at least 10 km per trial and a driving time of approximately 20 min. The tasks could be used to test the use of prior knowledge of varying strength, such as corridor width information.

In addition, calibration motions were recorded for both bikes. These were used to calibrate the position ${\vec{t}}_{I}$ and orientation $Q_{I}^{B}$ of the IMUs with respect to the body frame.

### 2.3. Observability Criteria

A systems state is observable if it can be determined uniquely in a given interval $\left[t_{0},t_{end}\right]$ given the observed output measurements *z* of the interval [27]. It is not required that the state is measured directly. The observability of nonlinear systems can be analyzed using the observability Grammian [12,28]. In general, the analysis is performed without taking noise into account.

We decided to use the weak observability of [29], which is based on the inverse function theorem. Any function can be locally inverted if its Jacobian is nonsingular; i.e., if it has full rank. A system is called weakly observable if the Jacobian of its measurement equations has full rank. Weakly observable means that the state can be determined uniquely around a working point. For example, given the sine output y(x)=sin(x), *x* is only weakly observable since the inversion $y^{-1}\left(y\left(x\right)\right)$ is ambiguous. The value of *x* is only unique if the region of *x*; e.g., x∈[−π,π] is given. In contrast, global observability means that the state can be determined everywhere in the possible state space. In practice, local observability is easier to prove, since it resolves in a Jacobian rank test.

The method requires measurement equations *Z* that depend on the *N* unknown states $x_{1}\cdots x_{N}$ and the *M* known, possibly time varying parameters $y_{1}\cdots y_{M}$. One measurement equation is required per unknown state. If there are less than *N* measurement equations available, further equations can be generated by derivation. Thus, if only one measurement equation *z* is available, N−1 derivatives have to be used. The Jacobian is formed with respect to the unknown states only:(16)$$J=\left(\begin{array}{ccc} \frac{\partial z}{\partial x_{1}} & \cdots & \frac{\partial z}{\partial x_{N}}\\ \vdots & \ddots & \vdots \\ \frac{\partial z^{\left(N-1\right)}}{\partial x_{1}} & \cdots & \frac{\partial z^{\left(N-1\right)}}{\partial x_{N}} \end{array}\right)$$

The system is weakly observable around a configuration $x_{1}\cdots x_{N}$, $y_{1}\cdots y_{M}$ if this Jacobian has full rank; i.e., if the determinant is nonzero. Another way to test the rank is to check linear independency of the rows or columns. Similarly to the observability Grammian, the unobservable substate is the nullspace of the Jacobian. Hence, a rank defect does not mean that all states are unobservable. The system remains with N−rank(J) unobservable degrees of freedom (DOFs).

In this paper, we depend on former observability proofs [12] which make *F* states $x_{K:N}$ (K=N−F+1) observable. Consequently, we can treat them as known parameters without having columns for them in the Jacobian, since their observability has been proven. To see that this is correct, consider the following: The measurement equations used depend on $x_{K:N}$. If we were to add them and their corresponding measurement equations $Z^{*}$ to the Jacobian, we would get a block shaped Jacobian of the form:(17)$$J^{*}=\left(\begin{array}{cc} J & \frac{\partial Z}{\partial x_{K:N}}\\ 0_{F:N} & \frac{\partial Z^{*}}{\partial x_{K:N}} \end{array}\right)$$
where $0_{F:N}$ is a F×N matrix of zeros. Due to the block shape, $J^{*}$ has full rank, if *J* and $\frac{\partial Z^{*}}{\partial x_{K:N}}$ have full rank. Since the states $x_{K:N}$ are observable, $\frac{\partial Z^{*}}{\partial x_{K:N}}$ has full rank. Hence, only the rank of *J* determines the observability of the unknown states.

## 3. Results of the Track Cycling Experiment

The trials are evaluated using the measurements of the main frame IMU. The laser barriers are only used as ground truth. The estimated trajectories stay on the track (see Figure 6).

Since we do not have continuous ground truth position, we evaluate the error indirectly. We predict when the center line of the bike passes the laser barriers and compare the prediction with the measurement (see Figure 6). Table 1 shows error metrics. To transfer the time error into a position error, multiply it by the highest velocity of the trial (∼12 m/s).

The estimator predicts passing the laser barrier with an average RMS of 0.090 s. The prediction error does not increase over time, but it has a random component. Since it does not increase, the estimate error is drift-free. It performs better than the previous estimator [11], which had an RMS of 0.096 s.

High time errors occur often at the first and last rounds (see Figure 6). Since we did not apply a start or end position constraint on the state estimate, the end and start point are less constrained than a point in the middle of the trial. Therefore, the error can be higher at those points. Additionally, the bikes are slower at the the start and end (roll out), which increases the dead reckoning time between the curves.

In Trial 3, the end point almost leaves the track. This is an additional hint that the constraints are weaker at end points.

The standard deviations are lower than the RMS. Thus, the estimate is biased. To have a measure of the bias, the mean error is used. The highest mean error is −0.176 s. In most cases, the bias is almost equal to the mean absolute error. Hence, it has a big impact on the quality of the estimate.

The average standard deviation decreased from 0.077 to 0.048 s compared to the previous estimator [11]. Hence, the precision of the estimator has greatly increased by changing the body frame.

The biases are negative for all trials and barriers. They stay similar over the trials. Barrier 1 has a outstanding high bias error. In our previous study, we claimed that the biases are caused by the prior knowledge [11]. Probably, they were caused by the old body frame position which falsifies the forward velocity prior. Now, the improved estimator reveals that the bias is likely to be systematic. Higher than expected alignment errors between the height map and the laser barriers may be a source of the bias.

In addition to the laser barriers, a video analysis has been performed for Trial 1. The estimated position of the bike is reprojected onto the video. This method is unsuitable for quantitative findings, but qualitative statements can be taken. It appears that the lateral error of the tracking (the error in the direction of the track) does increase at the straight parts in contrast to the curves.

Overall, the estimates are surprisingly close to the ground truth data, given that there is no exteroceptive sensor. The drift is corrected by the used prior knowledge. The average RMS of 0.090 s (∼1.08 m) is low and comparable to the accuracy of pedestrian tracking systems without dedicated external hardware [1,7]. The error has a few high peaks, which would affect an application. At the current state, the method cannot compete in accuracy with off-the-shelf indoor GNSS systems with ±2 cm error [30]. Nevertheless, we consider our results a successful tracking of the biker, which does not require external reference sensors.

## 4. Results of the Observability Analysis

The analysis of the observability properties is not straightforward. It is split into multiple steps (see Figure 7). It is already proven [12] that the body velocity ${\vec{v}}_{b}$, the roll φ and the pitch θ are observable with the forward velocity prior, if the vehicle drives on a curve. The same holds for the biases. These allow one to infer additional information, such as the yaw rate $\psi^{\prime }$. Based on the known states, the measurement equation $z^{\prime }$, which allows one to analyze the height-map prior, is inferred in Section 4.1. The height-map prior is analyzed for three surface types: tilted planes, cones and cone-like surfaces. The tilted planes and cone-like surfaces appear in the track of the Sixdays. It will be shown that driving curves on a tilted plane makes the yaw observable (Section 4.2), whereas driving on a cone-like surface can make the full state observable (Section 4.4 and Section 4.5). The analysis of the cone (Section 4.3) is performed as a specific case of the general cone-like surfaces. Additionally, the observability of general surfaces is analyzed for a given yaw (Section 4.6).

### 4.1. Measurement Equation of the Height-Map Prior

To apply the observability criterion of [29] the unknown DOFs have to be identified. The ultimate goal is to observe the 3D pose (position $\vec{p}$ and orientation *q*) of the tracked bike. The analysis is performed on the general motion model instead of the discretized model in Section 2.1:(18)$$\left(\begin{array}{c} {\dot{\vec{p}}}_{W}\\ {\dot{\vec{v}}}_{W}\\ {\dot{Q}}_{W}^{B} \end{array}\right)=\left(\begin{array}{c} {\vec{v}}_{W}\\ {\vec{a}}_{W}\\ Q_{W}^{B}*{\left[{\vec{\omega }}_{B}\right]}_{\times } \end{array}\right),$$
where ${\left[{\vec{\omega }}_{B}\right]}_{\times }$ is the skew symmetric matrix of ${\vec{\omega }}_{B}$. The body velocity is constrained to lay in forward direction; i.e.:(19)$${\vec{v}}_{b}=\left(\begin{array}{c} v_{f}\\ 0\\ 0 \end{array}\right),$$
where $v_{f}$ is the forward velocity. With this forward velocity prior [8], observability of the roll φ and pitch θ is already achieved when the bike drives on a curve. Additionally, IMU biases and the forward velocity become observable. Hence, the four DOFs of the position and the yaw ψ are unknown. From now on, all observability proofs are valid only if the bike drives a curve, since this is required for the observability induced by the forward velocity prior.

The position is constrained to lay on the track. Thus, we can define a function z(x,y) which maps x,y coordinates to the corresponding height as in [31]. This further reduces the number of unknown DOFs to three, because *z* can be calculated if *x* and *y* are known.

Possibly, the three unknown DOFs are observable through the height-map prior. Therefore, we designed our measurement equation based on the height function z(x,y). The height function can not be used as the measurement equation directly, since no measurement of the height *z* is available. Instead, its derivative is used. The value of $\dot{z}\left(t\right)$ can be calculated from the known pitch and forward velocity. We omit the function parameter (t) on the right side of the equations for simplicity:(20)$$\dot{z}\left(t\right)={\left(R_{z}\left(\psi \right)*R_{y}\left(\theta \right)*R_{x}\left(\varphi \right)*\left(\begin{array}{c} v_{f}\\ 0\\ 0 \end{array}\right)\right)}_{z}={\left(R_{z}\left(\psi \right)*R_{y}\left(\theta \right)*\left(\begin{array}{c} v_{f}\\ 0\\ 0 \end{array}\right)\right)}_{z}={\left(R_{y}\left(\theta \right)*\left(\begin{array}{c} v_{f}\\ 0\\ 0 \end{array}\right)\right)}_{z},$$
where $R_{v}\left(\alpha \right)$ creates a matrix which rotates a around the axis *v* by angle α. ${\left(\cdots \right)}_{z}$ retrieves the z component of a 3D vector. The yaw rotation has no influence, since we only retrieve the *z*-component. Thus, we can further reduce to:(21)$$\dot{z}\left(t\right)=v_{f}\cdot \sin\left(\theta \right)$$

The measurement model is then simply the derivative of z(x,y):(22)$$\dot{z}\left(t\right)=\dot{z}\left(x,y\right)=\frac{\partial z(x,y)}{\partial x}\cdot \dot{x}+\frac{\partial z(x,y)}{\partial y}\cdot \dot{y}$$

This introduces the two unknown functions $\dot{x}$ and $\dot{y}$ which are the world *x*-velocity $v_{x}$ and *y*-velocity $v_{y}$ respectively. These functions can be modeled with respect to the yaw and the planar velocity $v_{p}$ ( the velocity in the *x*-*y* plane):(23)$$\left(\begin{array}{c} v_{x}\\ v_{y} \end{array}\right)=v_{p}\cdot \left(\begin{array}{c} \cos(\psi )\\ \sin(\psi ) \end{array}\right)$$

The planar velocity itself can be retrieved from the forward velocity and the pitch:(24)$$v_{p}=\left|v_{x:y}\right|=\left|{\left(R_{z}\left(\psi \right)*R_{y}\left(\theta \right)*\left(\begin{array}{c} v_{f}\\ 0\\ 0 \end{array}\right)\right)}_{x:y}\right|=\left|{\left(R_{y}\left(\theta \right)*\left(\begin{array}{c} v_{f}\\ 0\\ 0 \end{array}\right)\right)}_{x:y}\right|$$

Since we use the norm of the $v_{x:y}$ vector it is independent of the yaw and can be further reduced to:(25)$$v_{p}=v_{f}\cdot \cos\left(\theta \right)$$

With those substitutions, the measurement equation only depends on known parameters and the three unknown states x, y and yaw:(26)$$\dot{z}\left(t\right)=\dot{z}\left(x,y,\psi \right)=\frac{\partial z(x,y)}{\partial x}\cdot v_{p}\cos\left(\psi \right)+\frac{\partial z(x,y)}{\partial y}\cdot v_{p}\sin\left(\psi \right)$$

This measurement equation effectively states that the vehicle has to stay on the surface. To clarify this, the equation can be reformed with respect to the surface normals. The surface normal can be calculated by:(27)$$N\left(x,y\right)=\left(\begin{array}{c} \frac{\partial z(x,y)}{\partial x}\\ \frac{\partial z(x,y)}{\partial y}\\ -1 \end{array}\right)$$

Rearranging the measurement equation yields:(28)$$0=\frac{\partial z(x,y)}{\partial x}\cdot v_{x}+\frac{\partial z(x,y)}{\partial y}\cdot v_{y}-1\cdot \dot{z}$$

This equation can be expressed by the dot product of the normal and the velocity vector with $\dot{z}=v_{z}$:(29)$$0=N\left(x,y\right)*\vec{v}\left(t\right)$$

Thus, the equation effectively states that the velocity vector has to be orthogonal to the surface normal, wherefore the trajectory cannot leave the surface.

The pose trajectory is effectively a 3D curve. It is often useful to define such a curve with regard to its arc length so that the norm of the derivative is one, which would equal:(30)$$v_{z}{\left(l\right)}^{2}+v_{p}{\left(l\right)}^{2}=1$$
where *l* is the arc length. In our case, it is more useful to parametrize the curve with regard to the planar distance traveled *s* so that:(31)$$v_{p}\left(s\right)=1$$

Following [32], the curve can be parametrized with respect to the planar traveled distance *s* since $\frac{\partial s}{\partial t}=v_{p}\left(t\right)\neq 0$ (we assume that we always have a planar motion).

Reparametrizing the measurement model is simple, but it is required that we are able to calculate the measurement value $\frac{\partial z(s)}{\partial s}=z^{\prime }\left(s\right)$ on the left hand side of the measurement equation.

We parametrize *t* with respect to the planar distance traveled as t(s). Now we get the equation:(32)$$z^{\prime }\left(s\right)=\frac{\partial z(t(s))}{\partial s}=\frac{\partial z(t(s))}{\partial t}\cdot \frac{\partial t(s)}{\partial s}$$
$\frac{\partial z(t(s))}{\partial t}$ is the known time derivative of z. Hence, only $\frac{\partial t(s)}{\partial s}$ is required to calculate the value of $z^{\prime }\left(s\right)$. Calculating the derivative of the position with respect to *s* yields:(33)$$\frac{\partial \vec{p}\left(t\left(s\right)\right)}{\partial s}=\frac{\partial \vec{p}\left(t\left(s\right)\right)}{\partial t}\cdot \frac{\partial t(s)}{\partial s}$$

Calculating $v_{p}\left(s\right)$ as the norm of the x and y derivative we get:(34)$$v_{p}\left(s\right)=\sqrt{{\left(\frac{\partial \vec{p}\left(t\left(s\right)\right)}{\partial t}\right)}_{x}^{2}+{\left(\frac{\partial \vec{p}\left(t\left(s\right)\right)}{\partial t}\right)}_{y}^{2}}\cdot \frac{\partial t(s)}{\partial s}=\sqrt{v_{x}{\left(t\left(s\right)\right)}^{2}+v_{y}{\left(t\left(s\right)\right)}^{2}}\cdot \frac{\partial t(s)}{\partial s}$$

Using (Equation 31) and rearranging yields:(35)$$\frac{\partial t(s)}{\partial s}=\frac{1}{\sqrt{v_{x}{\left(t\left(s\right)\right)}^{2}+v_{y}{\left(t\left(s\right)\right)}^{2}}}$$

The value of $\sqrt{v_{x}{\left(t\left(s\right)\right)}^{2}+v_{y}{\left(t\left(s\right)\right)}^{2}}$ is the planar velocity $v_{p}\left(t\right)$ in time space. Hence, we can calculate the value of $z^{\prime }\left(s\right)$ from our measurements, which allows the reparameterization with respect to *s*.

The reparameterization yields the new measurement equation:(36)$$z^{\prime }\left(s\right)=z^{\prime }\left(x\left(s\right),y\left(s\right),\psi \left(s\right)\right)=\frac{\partial z(x,y)}{\partial x}\cdot \cos\left(\psi \right)+\frac{\partial z(x,y)}{\partial y}\cdot \sin\left(\psi \right)$$

Since three states are unknown, three equations are required, which we retrieve by further differentiation:(37)$$\begin{array}{cc} z^{\prime \prime } & =\frac{\partial z^{\prime }}{\partial x}\cdot \cos\left(\psi \right)+\frac{\partial z^{\prime }}{\partial y}\cdot \sin\left(\psi \right)+\frac{\partial z^{\prime }}{\partial \psi }\cdot \psi^{\prime } \end{array}$$(38)$$\begin{array}{cc} z^{\prime \prime \prime } & =\frac{\partial z^{\prime \prime }}{\partial x}\cdot \cos\left(\psi \right)+\frac{\partial z^{\prime \prime }}{\partial y}\cdot \sin\left(\psi \right)+\frac{\partial z^{\prime \prime }}{\partial \psi }\cdot \psi^{\prime }+\frac{\partial z^{\prime \prime }}{\partial \psi^{\prime }}\cdot \psi^{\prime \prime } \end{array}$$

The two new variables $\psi^{\prime }$ and $\psi^{\prime \prime }$ appear in the measurement equations. However, these are known since $\psi^{\prime }$ can be calculated from the gyrometer measurements similarly to (Equation 20):(39)$$\psi^{\prime }\left(s\right)={\left(R_{y}\left(\theta \right)*R_{x}\left(\varphi \right)*\omega \right)}_{z}$$
where ω is the angular rate measurement of the gyrometer. The roll φ can not be ignored, since ω can be nonzero in its *x* and *y* directions. $\psi^{\prime \prime }$ is automatically available as the derivative of $\psi^{\prime }$.

To analyze the observability of the system, the rank of the Jacobian *J* with respect to the unknown states has to be retrieved:(40)$$\mathrm{rank}\left(J\right)=\mathrm{rank}\left(\begin{array}{ccc} \frac{\partial z^{\prime }}{\partial x} & \frac{\partial z^{\prime }}{\partial y} & \frac{\partial z^{\prime }}{\partial \psi }\\ \frac{\partial z^{\prime \prime }}{\partial x} & \frac{\partial z^{\prime \prime }}{\partial y} & \frac{\partial z^{\prime \prime }}{\partial \psi }\\ \frac{\partial z^{\prime \prime \prime }}{\partial x} & \frac{\partial z^{\prime \prime \prime }}{\partial y} & \frac{\partial z^{\prime \prime \prime }}{\partial \psi } \end{array}\right)$$

The rank of the Jacobian highly depends on the surface function z(x,y). Hence, it is difficult to find general statements about the observability conditions. Instead, we perform the analysis for typical surfaces that appear in the race track, to shed light on the overall situation. The two surfaces that appear are tilted planes and cone-like surfaces.

### 4.2. Analysis of Planar Surfaces

Horizontal planar surfaces (see Figure 8a) are the most simple surfaces a vehicle can drive on. The height of a vehicle driving on these surfaces is constant as used in [33]. While driving on these surfaces makes the z position observable (since it is constant), it does not give any information on the other three states. z(x,y) is constant for a horizontal planar surface. Thus, all its derivatives are zero, wherefore all entries of J are 0.

Tilted planar surfaces (see Figure 8b) are more interesting. Tilted surfaces can be represented without loss of generality with:(41)z(x,y)=a·y,a∈R∖{0},
by rotating the coordinate system accordingly. The measurement equations collapse to:(42)$$\begin{array}{cc} z^{\prime }\left(s\right) & =a\cdot \sin(\psi ) \end{array}$$(43)$$\begin{array}{cc} z^{\prime \prime }\left(s\right) & =a\cdot \cos\left(\psi \right)\psi^{\prime } \end{array}$$(44)$$\begin{array}{cc} z^{\prime \prime \prime }\left(s\right) & =a\cdot \left(\cos\left(\psi \right)\psi^{\prime \prime }-\sin\left(\psi \right)\psi^{\prime 2}\right) \end{array}$$

**Theorem** **1.**
*The yaw ψ of a vehicle driving on a tilted planar surface is observable if it drives a curve.*


**Proof.** The measurement Jacobian for a tilted planar surface is:
(45)$$J=\left(\begin{array}{ccc} 0 & 0 & a\cdot \cos(\psi )\\ 0 & 0 & -a\cdot \sin\left(\psi \right)\psi^{\prime }\\ 0 & 0 & -a\left(\sin\left(\psi \right)\psi^{\prime \prime }+\cos\left(\psi \right)\psi^{\prime 2}\right) \end{array}\right)$$The rank of this matrix is always one except if ψ∈π/2+Zπ and $\psi^{\prime \prime }=\psi^{\prime }=0$, which means following the direction vector $\vec{d}$ in Figure 8b permanently. This can not happen, if the vehicle drives a curve. The null space null(J) of the matrix is:
(46)$$v\in \mathrm{null}\left(J\right)\Rightarrow J*v=\vec{0}\Leftrightarrow v=\left(\begin{array}{c} x\\ y\\ 0 \end{array}\right),\;\;x,y\in R$$Thus, the yaw is observable. □

In practice, one may retrieve the yaw with the following equation:(47)$$\psi =\left\{\begin{array}{cc} \mathrm{atan}2\left(\frac{z^{\prime \prime }\left(s\right)}{\psi^{\prime }},z^{\prime }\left(s\right)\right)=\mathrm{atan}2\left(a\cdot \cos(\psi ),a\cdot \sin(\psi )\right) & \mathrm{if}\psi^{\prime }\neq 0\\ \mathrm{atan}2\left(\frac{z^{\prime \prime \prime }\left(s\right)}{\psi^{\prime \prime }},z^{\prime }\left(s\right)\right)=\mathrm{atan}2\left(a\cdot \cos(\psi ),a\cdot \sin(\psi )\right) & \mathrm{if}\psi^{\prime }=0,\psi^{\prime \prime }\neq 0 \end{array}\right.$$

Since *x* and *y* do not appear in the measurement equations (and all further derivatives), they are unobservable from the height-map prior.

### 4.3. Analysis of Cone Surfaces

A general cone (see Figure 9a) can be represented by:(48)z(x,y)=r(x,y)·m,m>0,
where r(x,y) is the planar distance to the center point and *m* is the inclination of the cone. Since cones are bodys of rotation, it is useful to use polar coordinates instead of Cartesian coordinates to analyze the cone using the substitutions:(49)$$\begin{array}{cc} x(s) & =r\cos(\phi ) \end{array}$$(50)$$\begin{array}{cc} y(s) & =r\sin(\phi ) \end{array}$$
where ϕ is the angle between the vector $\left(\begin{array}{c} x(s)\\ y(s) \end{array}\right)$ and the x axis.

The derivatives $r^{\prime }$ and $\phi^{\prime }$ can be calculated as follows:(51)$$\begin{array}{cc} r(s) & =\sqrt{x^{2}+y^{2}} \end{array}$$(52)$$\begin{array}{cc} r^{\prime }\left(s\right) & =\frac{xx^{\prime }}{\sqrt{x^{2}+y^{2}}}+\frac{yy^{\prime }}{\sqrt{x^{2}+y^{2}}} \end{array}$$

Using (Equation 23) with (Equation 31) yields:$$\begin{array}{ccc} r^{\prime }\left(s\right) & =\frac{r\cos(\phi )\cos(\psi )+r\sin(\phi )\sin(\psi )}{r} & \left(53\right)\\ \; & =\cos(\phi -\psi ) & \left(54\right) \end{array}$$

Similarly for $\phi^{\prime }\left(s\right)$:(55)$$\begin{array}{cc} \phi (s) & =\mathrm{atan}(y/x) \end{array}$$$$\begin{array}{ccc} \phi^{\prime }\left(s\right) & =\frac{xy^{\prime }}{x^{2}+y^{2}}-\frac{yx^{\prime }}{r^{2}+y^{2}} & \left(56\right)\\ \; & =\frac{r\cos(\phi )\sin(\psi )}{r^{2}}-\frac{r\sin(\phi )\cos(\psi )}{r^{2}} & \left(57\right)\\ \; & =-\frac{\sin(\phi -\psi )}{r} & \left(58\right) \end{array}$$

The measurement equation of the general cone follows:(59)$$\begin{array}{cc} z^{\prime }\left(s\right) & =r^{\prime }\cdot m=\cos\left(\phi -\psi \right)\cdot m \end{array}$$

Since a general cone is symmetric in the rotation around its center, it is impossible to observe either ϕ or ψ. Instead, only the difference of the two angles δ=ϕ−ψ can be observed. δ represents the angle of the vehicle direction ${\vec{d}}_{v}$ in relation to the maximum inclination direction ${\vec{d}}_{i}$ at the current position of the vehicle (see Figure 9a).

The distance to the cone center *r* does not appear in the measurement equation. But it appears in the derivative:$$\begin{array}{ccc} z^{\prime \prime }\left(s\right) & =\frac{z^{\prime }}{d\phi }\cdot \phi^{\prime }+\frac{z^{\prime }}{d\psi }\cdot \psi^{\prime } & \left(60\right)\\ \; & =m\cdot \left(\frac{\sin{\left(\phi -\psi \right)}^{2}}{r}+\sin\left(\phi -\psi \right)\psi^{\prime }\right) & \left(61\right) \end{array}$$

Hence, we can retrieve information about *r* and δ from the measurement.

**Theorem** **2.**
*The relative angle δ and the planar distance to the cones center r of a vehicle driving on a general cone surface are observable unless the vehicle drives parallel to the direction of maximum inclination.*


**Proof.** We substitute δ=ϕ−ψ into the measurement equations:
(62)$$\begin{array}{cc} z^{\prime }\left(s\right) & =\cos(\delta )\cdot m \end{array}$$
(63)$$\begin{array}{cc} z^{\prime \prime }\left(s\right) & =m\cdot \left(\frac{\sin{\left(\delta \right)}^{2}}{r}+\sin\left(\delta \right)\psi^{\prime }\right) \end{array}$$The Jacobian of $z^{\prime }\left(s\right)$ and $z^{\prime \prime }\left(s\right)$ with respect to the two unknown states δ and *r* is:
(64)$$J=\left(\begin{array}{cc} -\sin(\delta )\cdot m & 0\\ m\cdot \left(\frac{2\sin(\delta )\cos(\delta )}{r}+\cos\left(\delta \right)\psi^{\prime }\right) & -m\cdot \left(\frac{\sin{\left(\delta \right)}^{2}}{r^{2}}\right) \end{array}\right)$$
*J* has full rank if its determinant is nonzero. Hence, *r* and δ are observable unless the determinant is zero:
(65)$$\left|J\right|=m^{2}\cdot \frac{\sin{\left(\delta \right)}^{3}}{r^{2}}=0$$Thus, the Jacobian is not full ranked if sin(δ)=0; i.e.:
(66)δ∈ZπEquation (Equation 66) only holds if the vehicle travels in the direction of the maximum inclination ${\vec{d}}_{i}$ or in the exact opposite. Thus, the Jacobian has full rank and implies observability of *r* and δ if the travel direction is not parallel to the direction of maximum inclination. □

### 4.4. Analysis of Cone-Like Surfaces

The curves of the Sixdays track are not cone shaped. The inclination *m* of the curves is not constant, but changes depending on the angle ϕ. We call such a surface a cone-like surface (see Figure 9b).

**Definition** **1.**
*A cone-like surface is a surface which height function can be stated as:*
(67)$$z\left(r,\phi \right)=\left(r-r_{0}\right)\cdot m\left(\phi \right),$$
*where r>0 is the distance to the cone center, $r_{0}>0$ and m(ϕ)>0 is the inclination function.*


The parameter $r_{0}$ represents the radius at which the cone-like surface has 0 height. At this height, an *x*-*y* cut of the surface shows a perfect circle with radius $r_{0}$.

The dependency of *m* on ϕ complicates the analysis of the observability. Since ϕ appears without ψ, three equations are required. The base measurement equation changes to:(68)$$z^{\prime }\left(s\right)=r^{\prime }\cdot m\left(\phi \right)+\left(r-r_{0}\right)\cdot m^{\prime }\left(\phi \right)\phi^{\prime }=\cos\left(\phi -\psi \right)\cdot m\left(\phi \right)-\left(1-\frac{r_{0}}{r}\right)\cdot m^{\prime }\left(\phi \right)\sin\left(\phi -\psi \right)$$

The two derivatives can be found in Appendix B. Due to the complexity of the equations, only specific maneuvers can be analyzed reasonable, as it was done in [34]. It is important to note that in our case we do not know which maneuver is taken by the biker. Therefore, it is not allowed to simplify the measurement equations with the maneuver. The conditions of the maneuvers have to be substituted into the Jacobian instead. The MATLAB symbolic toolbox [35] has been used to handle the analysis of the observability (Matlab scripts available at: https://github.com/TomLKoller/ZaVI_TrackCycling/tree/master/MatlabProofs).

**Theorem** **3.**
*The angle ϕ of a vehicle driving on a cone-like surface is unobservable if the vehicle stays at a constant planar distance r=r0 to the center.*


**Proof.** Staying at a constant *r* implies that $r^{\prime }=0$, with (54):
(69)$$\begin{array}{cc} 0 & =r^{\prime }\left(s\right)=\cos\left(\phi -\psi \right) \end{array}$$
(70)$$\begin{array}{cc} \Rightarrow \psi & =\phi +\frac{\pi }{2}+Z\pi \end{array}$$Now using the derivative of ψ:
(71)$$\begin{array}{cc} \psi^{\prime }\left(s\right) & =\phi^{\prime }=-\frac{\sin(\phi -\psi )}{r}=-\frac{\sin(-\pi /2)}{r}=\frac{1}{r} \end{array}$$Since *r* is constant $\psi^{\prime }$ and $\phi^{\prime }$ are also constant.
(72)$$\begin{array}{cc} \Rightarrow \psi^{\prime \prime } & =\phi^{\prime \prime }=0 \end{array}$$Before we calculate the Jacobian of the system, we substitute δ=ϕ−ψ so that we can calculate the Jacobian with respect to ϕ,r and δ. The value of δ follows from the value of ψ:
(73)$$\delta =\phi -\psi =-\frac{\pi }{2}$$Now, we substitute the values of $\delta ,\psi^{\prime },\psi^{\prime \prime }$ and *r* into the observability Jacobian derived from the measurement equations (Appendix B):
(74)$$J=\left(\begin{array}{ccc} 0 & \frac{m^{\prime }\left(\phi \right)}{r_{0}} & m\left(\phi \right)\\ 0 & \frac{m^{\prime \prime }\left(\phi \right)}{r_{0}^{2}}-\frac{m\left(\phi \right)}{r_{0}^{2}} & \frac{2\;m^{\prime }\left(\phi \right)}{r_{0}}\\ 0 & \frac{m^{\prime \prime \prime }\left(\phi \right)}{r_{0}^{3}}-\frac{3\;m^{\prime }\left(\phi \right)}{r_{0}^{3}} & \frac{3\;m^{\prime \prime }\left(\phi \right)}{r_{0}^{2}}-\frac{m\left(\phi \right)}{r_{0}^{2}} \end{array}\right)$$Column 1 is the 0-vector, which means that the Jacobian is rank deficient. Since the system is nonlinear, additional derivatives of the measurement equation may yield nonzero elements in the first column. The first column corresponds to the change of the measurement equation in the direction of ϕ. Since the measurement equation is 0 for every ϕ given that r=r0, there can not be any nonzero entry. Thus, ϕ is completely unobservable from the height-map prior. □

The remaining unknown DOFs δ and *r* are observable, as long as the second and third column of the Jacobian are linearly independent.

### 4.5. Numerical Analysis of the Sixdays Track

Analysing the observability on cone-like surfaces beyond Theorem 3 is astonishing complex. The only possible way for us to retrieve meaningful statements about the observability was to calculate the determinants of the Jacobian numerically.

For numeric analysis, the inclination function m(ϕ) of the Sixdays track’s curves has been derived from the 3D model. The inclinations per angle ϕ were retrieved by least squares minimization and a fourth order polynomial was fitted onto the inclinations (see Figure 10):(75)$$m\left(\phi \right)=-0.0227\phi^{4}+0.133\phi^{3}-0.3351\phi^{2}+0.4124\phi +0.5775$$
$r_{0}$ has been set to 10 m.

Figure 11 shows the determinants of trajectories with constant radius *r*. The determinants are greater 0 for every $r>r_{0}$. Hence, the pose is completely observable for every movement on a constant radius with $r>r_{0}$.

Intuitively, we interpret the magnitude of the determinant as an indicator for the degree of observability. The magnitude increases with increasing radius. Hence, the pose is better observable if the biker is driving on the outer line of the track. The determinant decreases around $\phi =\frac{\pi }{2}$. At this point, the inclination function is almost flat, wherefore it is almost a general cone around the middle of the curve.

### 4.6. Analysis with Known Orientation

In many applications the yaw is observable. For example, the magnetometer of the IMU can be used to measure the yaw if the magnetic field is sufficiently stable [36]. According to Theorem 1, the yaw gets observable if the vehicle drives a curve on a tilted planar surface. Even in cases where the yaw is unobservable, it has only a small error for short time periods of gyrometer integration. Hence, it is worthwhile to analyze the impact of a known yaw on the observability.

At a general cone surface the known yaw gives full observability since *r* and δ are observable anyway. The missing angle ϕ can be calculated by:(76)ϕ=ψ+δ

**Theorem** **4.**
*If the yaw of a vehicle driving on a surface is known, the system is unobservable only if the gradients of of all derivatives of z(s) have the same direction.*


**Proof.** Since the yaw is known, only *x* and *y* are the unknown states. Since the system is nonlinear, we have to form the Jacobian from all derivatives of z(s) to prove that a state is unobservable. The resulting Jacobian for general surfaces is:
(77)$$J=\left(\begin{array}{cc} \frac{\partial z^{\left(1\right)}\left(s\right)}{\partial x} & \frac{\partial z^{\left(1\right)}\left(s\right)}{\partial y}\\ \vdots & \vdots \\ \frac{\partial z^{\left(K\right)}\left(s\right)}{\partial x} & \frac{\partial z^{\left(K\right)}\left(s\right)}{\partial y} \end{array}\right),\;\;K=\infty$$The system is unobservable if rank(J)<2, which is the case if all submatrices $J_{ij}$ formed by taking two different nonzero rows i,j of J have a zero determinant:
(78)$$|J_{ij}|=\frac{\partial z^{\left(i\right)}\left(s\right)}{\partial x}\cdot \frac{\partial z^{\left(j\right)}\left(s\right)}{\partial y}-\frac{\partial z^{\left(i\right)}\left(s\right)}{\partial y}\cdot \frac{\partial z^{\left(j\right)}\left(s\right)}{\partial x}=0$$Rearranging yields:
(79)$$\frac{\frac{\partial z^{\left(i\right)}\left(s\right)}{\partial x}}{\frac{\partial z^{\left(i\right)}\left(s\right)}{\partial y}}=\frac{\frac{\partial z^{\left(j\right)}\left(s\right)}{\partial x}}{\frac{\partial z^{\left(j\right)}\left(s\right)}{\partial y}}$$The gradients can be parametrized with respect to the direction of the gradients $\alpha_{i}$ and $\alpha_{j}$ so that:
(80)$$\begin{array}{cc} \frac{\partial z^{\left(i\right)}\left(s\right)}{\partial x} & =|\nabla z^{\left(i\right)}\left(s\right)|\cos\left(\alpha_{i}\right) \end{array}$$
(81)$$\begin{array}{cc} \frac{\partial z^{\left(i\right)}\left(s\right)}{\partial y} & =|\nabla z^{\left(i\right)}\left(s\right)|\sin\left(\alpha_{i}\right) \end{array}$$
(82)$$\begin{array}{cc} \frac{\partial z^{\left(j\right)}\left(s\right)}{\partial x} & =|\nabla z^{\left(j\right)}\left(s\right)|\cos\left(\alpha_{j}\right) \end{array}$$
(83)$$\begin{array}{cc} \frac{\partial z^{\left(j\right)}\left(s\right)}{\partial y} & =|\nabla z^{\left(j\right)}\left(s\right)|\sin\left(\alpha_{j}\right) \end{array}$$
where $|\nabla z^{\left(i\right)}\left(s\right)|$ and $|\nabla z^{\left(j\right)}\left(s\right)|$ are the norms of the gradients of $z^{\left(i\right)}\left(s\right)$ and $z^{\left(j\right)}\left(s\right)$ respectively. Now we substitute them into Equation (Equation 79):
(84)$$\begin{array}{cc} \frac{|z^{\left(i\right)}\left(s\right)|\cos\left(\alpha_{i}\right)}{|z^{\left(i\right)}\left(s\right)|\sin\left(\alpha_{i}\right)} & =\frac{|z^{\left(j\right)}\left(s\right)|\cos\left(\alpha_{j}\right)}{|z^{\left(j\right)}\left(s\right)|\sin\left(\alpha_{j}\right)} \end{array}$$
(85)$$\begin{array}{cc} \frac{\cos(\alpha_{i})}{\sin(\alpha_{i})} & =\frac{\cos(\alpha_{j})}{\sin(\alpha_{j})} \end{array}$$
(86)$$\begin{array}{cc} \alpha_{i} & =\alpha_{j}+Z\pi \end{array}$$In the case of an unobservable system, the direction angles of the gradient of each derivative have to be parallel to the gradient of each other derivative. Thus, all gradients of all derivatives need to have the same direction. □

While Theorem 4 is simple to prove, it is hard to grasp. Thus, we try to analyze it further. Since the gradients of $z^{\prime }$ and $z^{\prime \prime }$ have the same direction, they are linearly dependent:(87)$$\nabla z^{\prime }\left(s\right)=\lambda \nabla z^{\prime \prime }\left(s\right)$$

This means that the direction of the gradient of $z^{\prime }$ has to stay constant for the whole trajectory if the trajectory is unobservable. Unfortunately, this statement can not be used for surfaces without knowing the trajectory, since $z^{\prime }$ depends on x,y and yaw. It is possible only if the direction of $\nabla z^{\prime }\left(s\right)$ is independent of the yaw.

Surfaces that consist solely of aligned, parabolic points fulfil this condition. These surfaces are extrusions of a 3D curve (see Figure 12). The direction of $\nabla z^{\prime }\left(s\right)$ does not depend on the yaw. Thus, the position is always unobservable on such a surface.

The unobservable direction can be determined by finding the null space of the Jacobian:(88)$$J*\mathrm{null}\left(J\right)=\vec{0}$$

By using the angle representation we get:(89)$$\vec{n}\in \mathrm{null}\left(J\right)\Leftrightarrow \left(\begin{array}{c} 0\\ 0 \end{array}\right)=J*\vec{n}=\left(\begin{array}{cc} |z^{\prime }\left(s\right)|\cos\left(\alpha \right) & |z^{\prime }\left(s\right)|\sin\left(\alpha \right)\\ |z^{\prime \prime }\left(s\right)|\cos\left(\alpha \right) & |z^{\prime \prime }\left(s\right)|\sin\left(\alpha \right) \end{array}\right)*\left(\begin{array}{c} n_{x}\\ n_{y} \end{array}\right)$$

By solving for the null space we get:(90)$$\left(\begin{array}{c} n_{x}\\ n_{y} \end{array}\right)=\left(\begin{array}{c} -\sin(\alpha )\\ \;\cos(\alpha ) \end{array}\right)$$
which is the direction of the gradients of $z^{\prime }\left(s\right)$ and $z^{\prime \prime }\left(s\right)$ rotated by $90^{\circ }$. Hence, the null space is orthogonal to $\nabla z^{\prime }\left(s\right)$.

## 5. Discussion

In this work, a foundation has been laid to analyze the observability of vehicles driving on surfaces. A general analysis approach based on the criteria of [29], has been developed. The measurement Equation (Equation 36) can be used to analyze arbitrary surfaces. Unfortunately, the observability highly depends on the surface equation and the taken trajectory. Thus, no general analysis of the observability could be performed.

Analysis has been performed for surface types that occur in the race track. The track consists of two straight parts, which are tilted planar surfaces and two cone-like curves. The yaw of the bike is observable in a race according to Theorem 1, because the bikers never drive straight upwards the slope. A limitation of this statement is that the biker is required to drive a curve to observe the velocity by the forward velocity prior. If the biker is driving straight, the whole analysis is pointless, since the z velocity and planar velocity are unobservable. Adding a wheel odometer would greatly improve the observability of the yaw, since the body velocity would be observable even when driving straight.

The qualitative video analysis shows that the lateral error increases at the straight parts in contrast to the curves. This is not surprising since the straight parts do not contain any information about the position. It also indicates that there is positional information available in the curve.

It was not possible to retrieve a clear mathematical answer, whether or not the pose is observable in the curves. Bikers tend to drive on a constant radius in a race to reduce the traveled distance. Hence, the pose would be observable according to our numerical analysis. However, Theorem 3 states that the pose is unobservable if the biker drives on the most inner line of the track. Since they tend to drive the shortest route, they drive close to the inner line. Thus, the pose may be poorly observable.

The drift-free experimental results with an RMS of 0.090 s (approximately 1.08 m) indicate that the pose is overall observable. Since the yaw drifts slowly, a few time periods with observable yaw (e.g., driving a curve on the straight parts) may be enough to retrieve a sufficient estimate of the yaw. This estimate may be enough to retrieve the pose in the curves even if the biker drives on the inner line.

The analysis revealed a weakness of the estimator. A trajectory, where the driver drives straight on the straight parts and on the inner line of the curves, may have insufficient observability of the pose. Therefore, an experiment with this kind of trajectory is required to verify the error bounds of the systems. In addition, the observability for trajectories on non-constant radii in the curves has to be examined experimentally, since it was not possible to calculate it analytically.

Overall, the observability analysis revealed some of the error properties of a bike driving on the race track. The height map prior can give observability of the pose but it is not guaranteed to do so. The drift-free estimates of the experiment are now better understood and ongoing development of the state estimator can benefit from the gained insight.

The general capability of the system to deliver drift free estimates of the pose has been proven. Thus, future work has to focus on improving the estimate until it reaches a usable level. An absolute reference system is required to quantify the true estimate error in all driving conditions. As a first improvement, the IMU should be positioned at the center of the rear wheel, since the transformation of the IMU measurements amplifies the IMU noise.

## Figures and Tables

**Figure 1 sensors-20-02438-f001:**
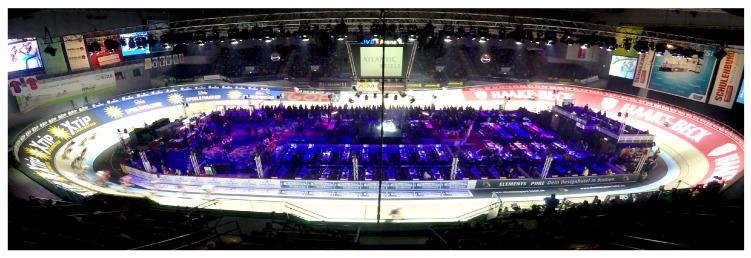
Ref. [11] The track of the Sixdays Bremen.

**Figure 2 sensors-20-02438-f002:**
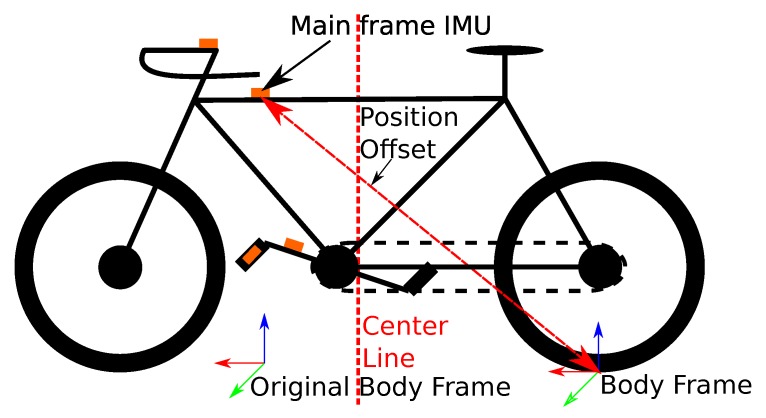
Bike with inertial measurement units (IMUs). Only the main frame IMU is used.

**Figure 3 sensors-20-02438-f003:**
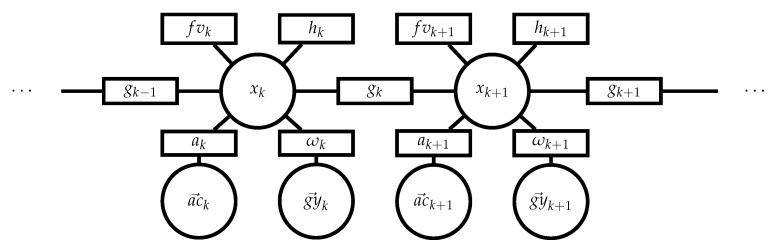
Ref. [11] The optimization problem as a factor graph. ${\vec{ac}}_{k}$ and ${\vec{gy}}_{k}$ are the inputs, $x_{k}$ the state, $g_{k}$ the dynamic model (Equation 8), $a_{k}$ the acceleration function (Equation 7), $\omega_{k}$ the angular rate function (Equation 6), $fv_{k}$ the forward velocity prior (Equation 9) and $h_{k}$ the height-map prior (Equation 10).

**Figure 4 sensors-20-02438-f004:**
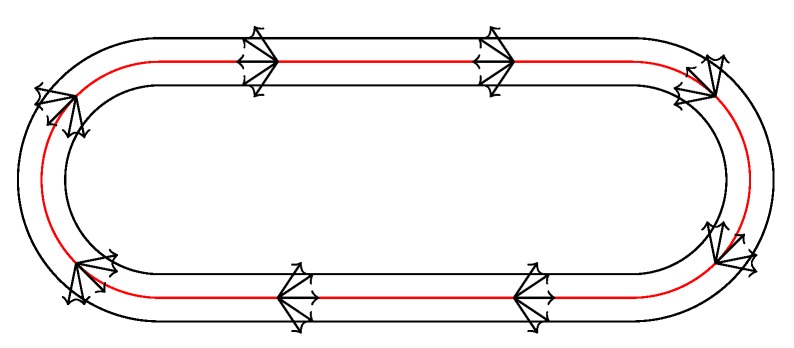
[11] Basic form of the track. The red line shows the 1D simplification. The arrows show the possible heading range.

**Figure 5 sensors-20-02438-f005:**
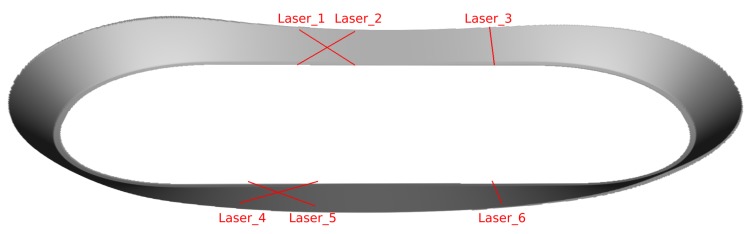
Ref. [11] Height map of the Sixdays track. The red lines show the laser barriers.

**Figure 6 sensors-20-02438-f006:**
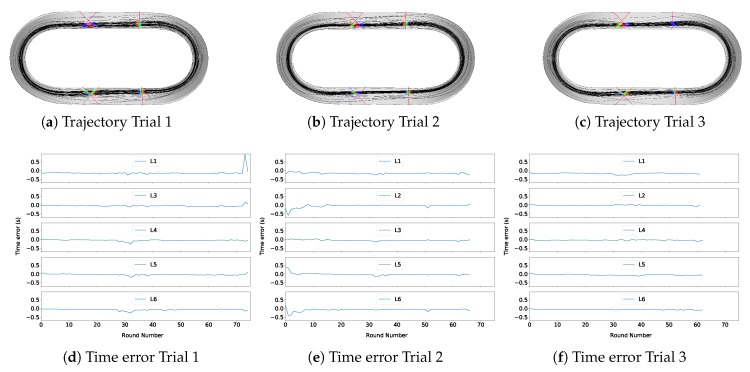
(**a**–**c**) Trajectory of a biker in all trials after 100 iterations. The lasers are shown as red lines. Blue color means that the estimator predicts that the bike passes a laser barrier. Red color means that a barrier measured that a bikes passes. Green color shows where prediction and measurement agree. (**d**–**f**) Time error between measured pass time and predicted pass time of the laser barriers.

**Figure 7 sensors-20-02438-f007:**
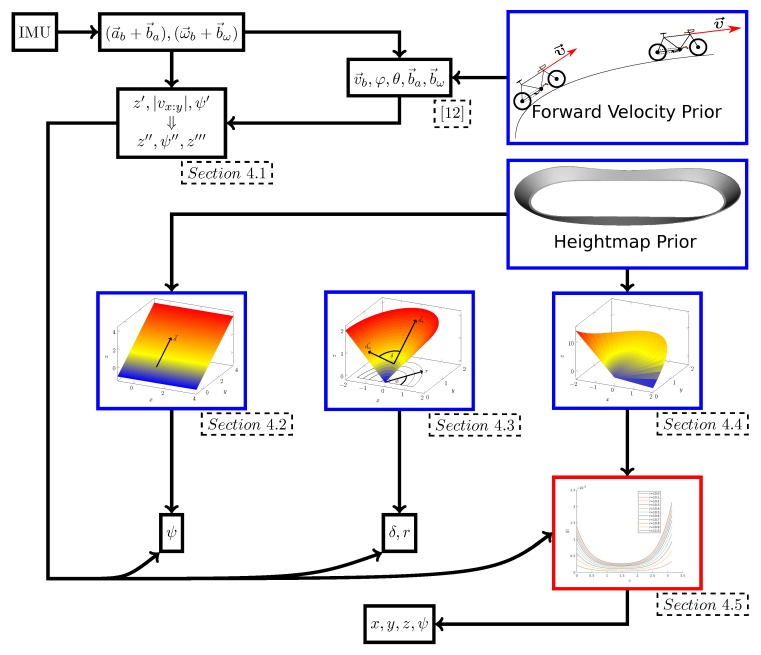
Overview of the observability analysis. The available information is arranged in the boxes. The arrows show which information is required to infer additional information. Blue boxes show the prior knowledge. Information in the red box is retrieved by numeric analysis, whereas all other information is derived analytically.

**Figure 8 sensors-20-02438-f008:**
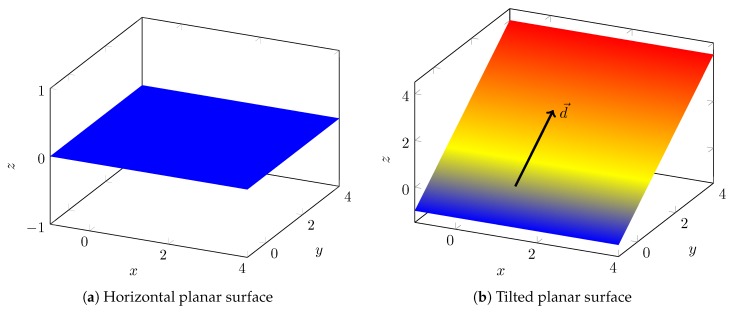
Planar surfaces.

**Figure 9 sensors-20-02438-f009:**
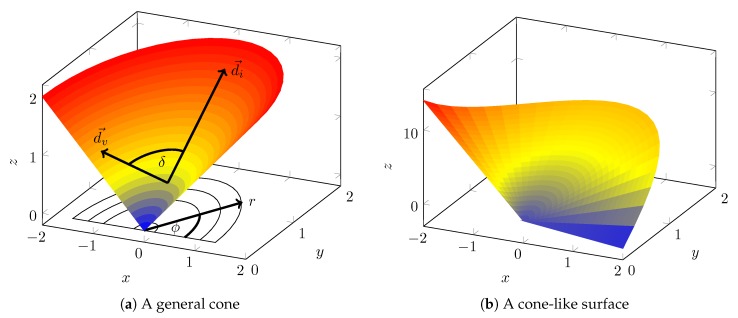
Cone-shaped surfaces.

**Figure 10 sensors-20-02438-f010:**
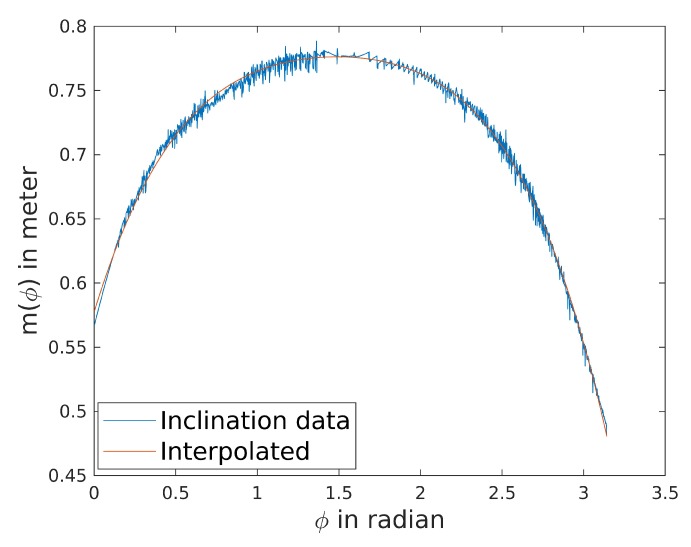
Inclination function of the Sixdays track.

**Figure 11 sensors-20-02438-f011:**
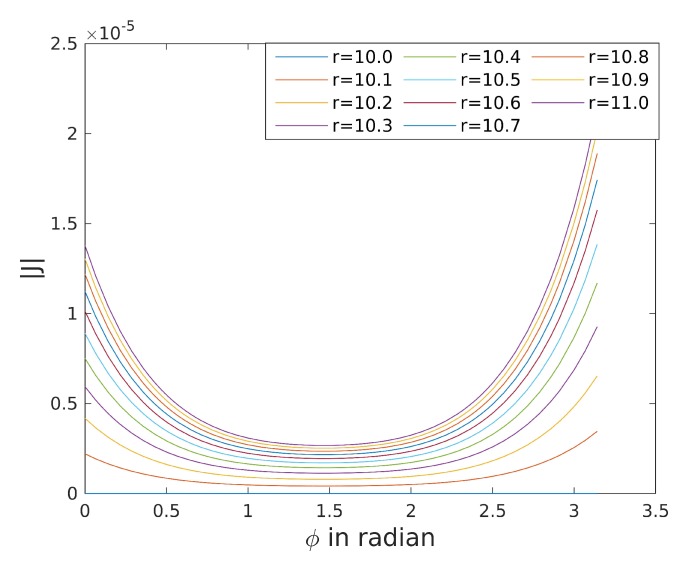
Determinant of the observability Jacobian for different radii on the Sixdays track with $r_{0}=10$.

**Figure 12 sensors-20-02438-f012:**
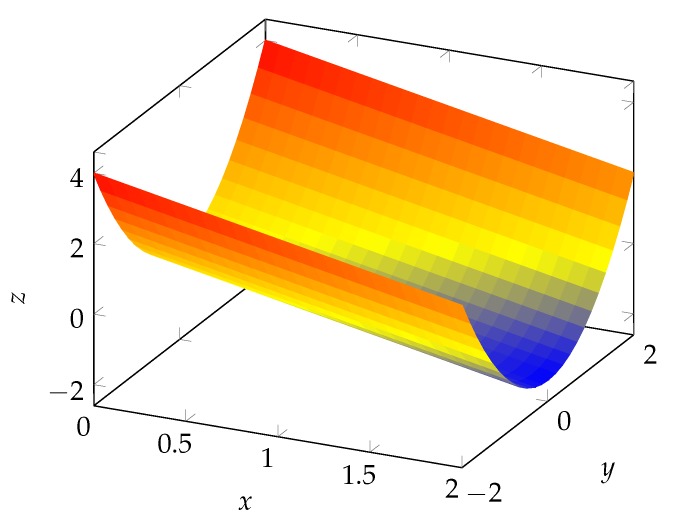
A surface with only parabolic points with the same direction for $k_{2}$.

**Table 1 sensors-20-02438-t001:** Mean error $\bar{e}$, mean absolute error $\bar{\left|e\right|}$, root-mean-squared error RMS, maximum error $e_{max}$ and standard deviation σ of time errors in seconds. Trial 1 (orange), Trial 2 (green) and Trial 3 (yellow).

Laser #	$\bar{e}$	$\bar{e}$	RMS	$e_{\max}$	σ	$\bar{e}$	$\bar{e}$	RMS	$e_{max}$	σ	$\bar{e}$	$\bar{e}$	RMS	$e_{\max}$	σ
1	−0.141	0.166	0.190	0.925	0.128	−0.159	0.159	0.164	−0.256	0.040	−0.176	0.176	0.180	−0.278	0.036
2	-	-	-	-	-	−0.050	0.053	0.103	−0.600	0.090	−0.017	0.023	0.027	−0.082	0.021
3	−0.047	0.053	0.061	0.163	0.039	−0.049	0.053	0.060	−0.129	0.034	-	-	-	-	-
4	−0.047	0.048	0.061	−0.267	0.039	-	-	-	-	-	−0.040	0.040	0.045	−0.114	0.021
5	−0.038	0.044	0.054	−0.190	0.038	−0.021	0.048	0.078	0.366	0.075	−0.052	0.054	0.060	−0.136	0.031
6	−0.074	0.074	0.085	−0.274	0.041	−0.077	0.084	0.109	−0.399	0.076	−0.069	0.069	0.071	−0.104	0.016

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
