# Peer review of "State Observability through Prior Knowledge: Analysis of the Height Map Prior for Track Cyclingâ€"

_sensors, 2020, doi:10.3390/s20092438_

Round 1
Reviewer 1 Report
This paper extends previous results on the use of the only inertial sensors for navigation, by exploiting kinematic motion constraints and a priori knowledge. In particular, the paper provides two new contributions:
1) A trajectory estimator validated by experiments to track bikers.
2) A follow up of the observability analysis in [12] to also account the knowledge of a height map.
Remarks/Comments.
In section 2 it is unclear why 2 biases are used for the accelerometer and 2 for the gyro-meter. An explanation should be added.
2) In section 4, the observability analysis does not follow the standard procedure based on the observability rank condition (i.e., based on the computation of the Lie derivatives). In particular, the same problem can be solved by characterizing the system with the state in (2), without the acceleration, the angular speed and the angular acceleration, and by considering the acceleration and the angular speed as 6 independent system inputs. In other words, we have:
\dot{X} = f_0(X) + \sum_{i=1}^6 f_i(X) u_i
where u_1,…,u_6 are the components of the acceleration and the angular speed in the IMU frame. The output functions are:
- the analytical expression of the speed orthogonal to the forward speed, in terms of the state components
- The function z-z(x,y)
By iteratively computing the Lie derivatives of the above outputs along f_1,…,f_6, we can obtain the observable codistribution.
This analysis could be simplified by using the results proved in [12]. This is obtained by adding also the three outputs:
- forward speed
- roll
- pitch
and by removing all the biases. Then we proceed as above, i.e., by computing the Lie derivatives of all the outputs along f_1,…,f_6
The advantages of the application of the Observability Rank Condition are:
- The derivation is automatic. It is unnecessary to introduce geometric and ad-hoc procedures to obtain the observability. The authors spent many pages to obtain their results: they could be obtained automatically.
- The analysis is more complete. To this regard, I want to remark that, while the results of the paper are correct when they prove that a given state is observable, they are actually not correct when they prove that a given state is not observable. I mean, the result could be correct (and I believe that it is) but not the proof. Indeed, proving that a given Jacobian is not full rank, does not suffice. It could be possible that, by adding a further time derivative, we can obtain a full rank Jacobian. Instead, the observability rank condition computes the observable codistribution by a convergent algorithm.
- I believe that, by using the observability rank condition, it is possible to obtain more general results.
On the other hand, I do not ask the authors to perform this analysis if they are not expert in the use of the observability rank condition. Unfortunately, as most of the results in the literature of control theory, it is introduced by paying attention to many insignificant details and its application seems something awful (in the framework of control theory, it has only been applied to toy-models). And this is really a pity since it is a very powerful tool that can be easily applied automatically (e.g., by using the symbolic tool of MATLAB). Note that in the framework of robotics, and in particular in visual-inertial navigation, the observability rank condition has been successfully applied to obtain the system observability of very complex systems.
Nevertheless, I would appreciate a math summary, by giving the equations for the dynamics and all the constraints. At a first read of the paper I was not happy because I believed that the observability analysis was based on the dynamics given by equation (8). This equation assumes that the time derivatives of the acceleration and of the angular acceleration vanish. This assumption is not motivated and it is not suitable for the observability analysis. Then, by reading the paper again, I realized that the observability analysis is not based on this assumption.
Maybe I find the title a bit inappropriate since it seems that the paper provides contributions in the framework of observability. This is not the case. The paper solves a very specific problem by using a very ad-hoc procedure.TYPOS/MINOR
Line 29 track track
Author Response
We thank the reviewer for his excessive comments. The observability rank condition looks promising. We are grateful that the reviewer invested his time to explain the rough idea of this methodoly.
In section 2 it is unclear why 2 biases are used for the accelerometer and 2 for the gyro-meter. An explanation should be added.
We added an explanation that we use the 2 bias model following [18], which claims to be more precise.
2) In section 4, the observability analysis does not follow the standard procedure based on the observability rank condition (i.e., based on the computation of the Lie derivatives). ...
We are looking forward to test this methodology in future publications.
- The analysis is more complete. To this regard, I want to remark that, while the results of the paper are correct when they prove that a given state is observable, they are actually not correct when they prove that a given state is not observable. I mean, the result could be correct (and I believe that it is) but not the proof. Indeed, proving that a given Jacobian is not full rank, does not suffice. It could be possible that, by adding a further time derivative, we can obtain a full rank Jacobian. Instead, the observability rank condition computes the observable codistribution by a convergent algorithm.
The reviewer is right. The proofs are incomplete. Thus, we applied the following changes:
Theorem 1:
-Moved the statement about unobservability of x and y outside of the proof and added that further derivatives also do not contain x,y.
Theorem 3:
-Added a statement about higher derivatives to validate the proof.
Theorem 4:
-Refactored the proof to take all derivatives into account.
Nevertheless, I would appreciate a math summary, by giving the equations for the dynamics and all the constraints.
The dynamic model and the forward velocity prior have been added to section 4.1.
Maybe I find the title a bit inappropriate since it seems that the paper provides contributions in the framework of observability. This is not the case. The paper solves a very specific problem by using a very ad-hoc procedure.
The title is meant to advertise the idea of achieving state observability by using prior knowledge instead of sensors. We already used the preamble in our previous publications. Thus, we would like to keep the title to keep the connection between the papers. Instead, we tried to clarify the purpose of the paper in the abstract, to prevent confusion with a contribution in the field of observability theory.TYPOS/MINOR
Line 29 track track
Corrected.
Reviewer 2 Report
This article aims to analyse the observability of the pose in the conducted experiment in order to improve the pose estimation of the previous studies.
The English language is quite adequate and the content is complete.
There are only few minor changes that I suggest to consider:
- Line 19: please change “GPS” with “GNSS” (and cite the acronym)
- Line 42: again, please change “GPS” with “GNSS”
- Line 95: please change “Formula” with “equation”
- Figure 6 -d -e -f: please use the same scale bar for these three graphs
- Line 316: please check the symbol at the end of the sentence
- Line 358: please check the symbol at the end of the sentence
The reference section is adequate, even if I suggest to extend the literature review considering also the following contribution, already published on Sensors journal: Gonzalez et al. (2019). Performance Assessment of an Ultra Low-Cost Inertial Measurement Unit for Ground Vehicle Navigation. Sensors, 19(18), 3865.
Even if the quality of the paper is very good, starting from the previous considerations, I can affirm that the paper is not ready to be accepted for publication in the present form but needs a minor revision.
Author Response
We thank you for your review, language and style hints.
- Line 19: please change “GPS” with “GNSS” (and cite the acronym)
- Line 42: again, please change “GPS” with “GNSS”
- Line 95: please change “Formula” with “equation”
Yes!
- Figure 6 -d -e -f: please use the same scale bar for these three graphs
We do, but it reduces visibilty of the smaller errors since, a few high values appear in the errors. However, the comparability is higher.
- Line 316: please check the symbol at the end of the sentence
- Line 358: please check the symbol at the end of the sentence
This is the symbol which is set by the proof environment of the tex class. It appears at the end of every proof (Everything that starts as Proof. ).
The reference section is adequate, even if I suggest to extend the literature review considering also the following contribution, already published on Sensors journal: Gonzalez et al. (2019). Performance Assessment of an Ultra Low-Cost Inertial Measurement Unit for Ground Vehicle Navigation. Sensors, 19(18), 3865.
The suggested paper compares a low cost IMU with a medium cost IMU in IMU/GNSS navigation and is relevant in the field of automotive driving. It does not focus on indoor positioning, prior knowledge or observability, which are the main parts of our manuscript. Hence, we would like to not include it in the literature review.